# *Aedes albopictus* Autophagy-Related Gene 8 (*AaAtg8*) Is Required to Confer Anti-Bacterial Gut Immunity

**DOI:** 10.3390/ijms21082944

**Published:** 2020-04-22

**Authors:** Chang Eun Kim, Ki Beom Park, Hye Jin Ko, Maryam Keshavarz, Young Min Bae, BoBae Kim, Bharat Bhusan Patnaik, Ho Am Jang, Yong Seok Lee, Yeon Soo Han, Yong Hun Jo

**Affiliations:** 1Department of Applied Biology, Institute of Environmentally-Friendly Agriculture (IEFA), College of Agriculture and Life Sciences, Chonnam National University, Gwangju 61186, Korea; chang9278@naver.com (C.E.K.); misson112@naver.com (K.B.P.); hjngo0129@naver.com (H.J.K.); Mariakeshavarz1990@gmail.com (M.K.); ugisaka@naver.com (Y.M.B.); kbb941013@gmail.com (B.K.); hoamjang@gmail.com (H.A.J.); 2P. G Department of Biosciences and Biotechnology, Fakir Mohan University, Vyasa Vihar, Nuapadhi, Balasore, Odisha 756089, India; drbharatbhusan4@gmail.com; 3School of Biotech Sciences, Trident Academy of Creative Technology (TACT), Chandrasekharpur, Bhubaneswar, Odisha 751024, India; 4Department of Biotechnology and Life Science, College of Natural Sciences, Soonchunhyang University, 22 Soonchunhyangro, Shinchang-myeon, Asan, Chungchungnam-do 31538, Korea; yslee@sch.ac.kr

**Keywords:** *Aedes albopictus*, autophagy, microbial infection, *AaAtg8*, RNAi, gut and abdominal carcass

## Abstract

Autophagy is an important process by which pathogens and damaged or unused organelles are eliminated. The role of autophagy in development and the immune response to pathogens is well established. Autophagy-related protein 8 (Atg8) is involved in the formation of the autophagosome and, with the help of the serine protease Atg4, mediates the delivery of both vesicles and the autophagosome to the vacuole. Here, we cloned the *Aedes albopictus* autophagy-related protein 8 (*AaAtg8*) gene and characterized its role in the innate immunity of the mosquito against microbial infections. *AaAtg8* is comprised of an open reading frame (ORF) region of 357 bp encoding a polypeptide of 118 amino acid residues. A domain analysis of *AaAtg8* revealed an Atg8 ubiquitin-like domain, Atg7/Atg4 interaction sites, and peptide binding sites. The *AaAtg8* mRNA expression was high in the Malpighian tubules and heads of both sugar-fed and blood-fed adult female mosquitoes. The expression level of *AaAtg8* mRNA increased in the midgut and abdominal carcass following being challenged with *Listeria monocytogenes*. To investigate the role of *AaAtg8* in the innate immune responses of *Ae. albopictus*, *AaAtg8* gene-silenced adult mosquitoes were challenged by injection or by being fed microorganisms in blood. High mortality rates were observed in mosquitoes in which *AaAtg8* was silenced after challenges of microorganisms to the host by blood feeding. This suggests that Atg8-autophagy plays a critical role in the gut immunity in *Ae. albopictus*.

## 1. Introduction

Autophagy is a conserved mechanism among metazoans, wherein dysfunctional cellular components are degraded in-bulk by the lysosome. Autophagy allows cells to recycle cellular components under conditions of stress, starvation, development, and cancer [1,2,3]. The machinery of autophagy has been implicated in inflammatory pathologic conditions primarily as a result of its function in various stress response mechanisms. Furthermore, defects in the autophagy pathway have been implicated in the etiology and progression of neurodegenerative disorders, including Alzheimer’s disease, Huntington’s disease, amyotrophic lateral sclerosis, and Parkinson’s disease. Many infection models have also considered the autophagy signaling pathway responsible for targeting pathogens for eventual degradation in autolysosomes [4,5]. Autophagy is classified into three types: microautophagy (non-selective autophagy), macroautophagy (selective autophagy), and chaperone-mediated autophagy (restricted to mammalian cells). In macroautophagy, an autophagosome comprising the sequestered cytoplasm and dysfunctional organelles is formed. This autophagosome subsequently fuses with the vacuolar membrane and discards the autophagic contents. Finally, the autophagic contents are degraded by hydrolase enzymes in the vacuole [6,7].

The autophagy mechanism was first discovered through the genetic screening of yeast (*Saccharomyces cerevisiae*) and is currently understood to contain 36 discrete genes [8]. Many of the autophagy genes have orthologs in vertebrates. These genes have been categorized into functional categories, such as “Atg1 kinase and its regulators”, “PtdIns 3-kinase complex”, “Atg12 conjugation system”, “Atg8 conjugation system”, “Atg2-Atg18 complex”, “Atg9”, “cytoplasm to vacuole targeting (Cvt) pathway”, “Pexophagy (selective peroxisomes degradation by autophagy)”, “Mitophagy (selective mitochondria degradation by autophagy)”, “Autophagic degradation and recycle”, and “others”. The Atg1 kinase complex (interactome of the Atg1 complex) interacts with Atg13 for optimal Atg1 kinase activity. In this interactome, the Atg17-Atg31-Atg29 sub-complex also sustains the Atg1 kinase activity. The phosphorylation of Atg13 and Atg1 by upstream nutrient sensors, such as target of rapamycin (TOR) and protein kinase A (PKA), induces autophagy [9]. The phosphatidylinositol (PtdIns) 3-kinase complexes (complexes I and II) formed of vacuolar protein sorting 15 (Vps15), Vps34, and Vps30/Atg6 regulate the recruitment of proteins that form the eventual autophagosome [10]. The transmembrane protein Atg9 is found in the periphery of the growing phagophore and is considered to be an essential regulator of autophagy initiation [11]. Finally, the autophagy mechanism includes two ubiquitin-like protein conjugation complexes: the Atg8-phosphatidylethanolamine (PE) complex (which includes the Atg8-modifying protease Atg4) and the Atg12-Atg5-Atg16 complex (which includes the activating enzyme Atg7 and two analogs of ubiquitin-conjugating enzymes, Atg3 and Atg10) [12]. Atg8 participating in autophagy is covalently linked to PE in another ubiquitination-like reaction that is mediated by Atg7 and Atg3 [13,14]. Earlier reports also suggest that Atg8 regulates the size of the autophagosome in selective autophagy [8,15]. In this context, the Atg8 protein is considered a reliable biomarker for autophagy induction and progression [16,17]. Furthermore, Atg8 is utilized in the Cvt pathway, mitophagy, and pexophagy in cargo binding [18]. The conservation of function among mammalian and insect homologs of Atg8 has been reported. The *Bombyx mori* Atg8 (*BmAtg8*) is highly expressed in many tissues and is reported to have both a high sequence conservation and a shared C-terminal ubiquitin-fold domain across species [19]. In *Spodoptera litura* cells, Atg8 is found localized in both the nucleus and the cytoplasm, and cytoplasmic localization occurs during the autophagic process [20]. In *Galleria mellonella*, Atg8 expression is positively correlated with starvation [21]. In the coleopteran pest, *Tenebrio molitor*, Atg8 is implicated in the process of the autophagy-mediated clearance of the intracellular pathogen *Listeria monocytogenes* [5]. Most importantly, *Aedes aegypti* Atg8 is known to play a critical role in the progression of gonadotrophic cycles [22]. In this study, we identified the Atg8 homolog in *Aedes albopictus* (*AaAtg8*) and assessed its immune functions during microbial infection in the host. We showed, through an RNA interference-based silencing assay, that *AaAtg8*-depleted adult mosquitoes are more susceptible to *L. monocytogenes* infection. Meanwhile, increased susceptibility against bacterial feeding (*Escherichia coli*, *Staphylococcus aureus*, *L. monocytogenes* and *Candida albicans*) was observed in the ds*AaAtg8*-treated group, suggesting that autophagy plays a critical role in antibacterial gut immunity in *Ae. albopictus*. 

## 2. Results

### 2.1. Sequence Analysis of AaAtg8

The *AaAtg8* open reading frame (ORF) sequence was identified by performing a local translated Basic Local Alignment Search Tool (TBLASTN) analysis using the *T. molitor* Atg8 (*Tm*Atg8) amino acid sequence (AJE26300.1) as a query against the *Ae. albopictus* RNA-sequencing database. The *AaAtg8* ORF sequence was 357 bp in length and encoded a polypeptide of 118 amino acid residues (Figure 1). Domain analysis of the *AaAtg8* protein sequence, completed using InterProScan5 and BLASTP, showed a characteristic Atg8 ubiquitin-like domain (amino acid residues 3 to 116). Additionally, the Atg8 ubiquitin-like domain of the *AaAtg8* protein contained an Atg7 interaction site, an Atg4 interaction site, and a peptide binding site (Figure 1). The highly-conserved lysine residue (K66), crucial to chain linkage with ubiquitin, was also found. 

The pairwise alignment of the *AaAtg8* nucleotide sequence from the Korean and Chinese strains (GenBank: MH243747.1) of *Ae. albopictus* revealed highly conserved sequences differing at only one nucleotide residue (A_216_ in *AaAtg8-*Korean strain and T_216_ in *AaAtg8*-Chinese strain) (Figure 2A). The protein sequences of *Aa*Atg8 from each strain were found to be identical (Figure 2B). 

### 2.2. Developmental and Tissue Distribution of AaAtg8 at mRNA Level

Using quantitative real-time PCR (qRT-PCR), we investigated the expression patterns of *AaAtg8* mRNA at ten different developmental stages and in five and six different tissues of *Ae. albopictus* sugar-fed and blood-fed adults, respectively (Figure 3). *AaAtg8* mRNA was found to be highly expressed in the larval stages of development, however the expression was low in the adult male stage (Figure 3A). For a tissue-specific expression analysis, we collected samples from sugar-fed and blood-fed adult female mosquitoes. In sugar-fed adult female mosquitoes, *AaAtg8* mRNA levels were highest in the Malpighian tubules (Figure 3B). This expression pattern differed from that of blood-fed adult female mosquitoes, for which *AaAtg8* mRNA levels were highest in the head (Figure 3C). 

### 2.3. Expression of AaAtg8 mRNA after Immune Challenge 

We investigated the time-dependent induction patterns of *AaAtg8* mRNA in the abdominal carcass and midgut tissues of *Ae. albopictus* adult mosquitoes after microorganism challenges. *AaAtg8* mRNA levels were analyzed at intervals of 3, 6, 9, and 12 h after challenges with bacterial (*E. coli*, *S. aureus*, *L. monocytogenes*) and fungal (*C. albicans*) pathogens (Figure 4). *AaAtg8* mRNA levels were highest in the abdominal carcass of sugar-fed mosquitoes following *L. monocytogenes* challenge (Figure 4A). Challenge with the gram-positive bacteria *S. aureus* also led to an increase in *AaAtg8* mRNA levels at 6 h post-challenge in the abdominal carcass of sugar-fed mosquitoes. In blood-fed mosquitoes, *AaAtg8* mRNA levels in the abdominal carcass tissue were dramatically increased at all time-points following the *L. monocytogenes* challenge, with lesser increases noted in mosquitoes challenged with other pathogens (Figure 4B). In mosquitoes challenged by artificial blood feeding with microorganisms, no significant differences in *AaAtg8* mRNA levels were observed in the abdominal carcass (Figure 4C). In the midgut tissue of sugar- and blood-fed mosquitoes (Figure 4D,E), *AaAtg8* mRNA levels were not significantly higher compared to the abdominal carcass tissue (Figure 4A,B) after being challenged with microorganisms. In the case of mosquitoes challenged by artificial blood feeding with microorganisms, *AaAtg8* mRNA levels in the midgut were significantly increased at 6 h and 12 h post-infection by the gram-negative bacteria *E. coli* and fungus *C. albicans* (Figure 4F). 

### 2.4. Effects of AaAtg8 Gene Silencing on the Survivability of Ae. Albopictus Mosquitoes

To functionally characterize *AaAtg8*, we silenced its expression in *Ae. albopictus* adults using RNAi (injection of ds*AaAtg8*) and monitored the host ability to resist infection. After confirmation of the *AaAtg8* mRNA knockdown efficiency (about 95%) (Figure 5A), we challenged the *AaAtg8*-silenced mosquitoes with systemic injection of *E. coli*, *S. aureus*, *C. albicans*, and *L. monocytogenes*. The mortality of *Ae. albopictus* adults (ds*AaAtg8* and ds*EGFP* treated groups) in response to the microbial challenge was monitored for a duration of 9 days. No significant differences in mortality were observed between the ds*AaAtg8* and ds*EGFP*-treated groups after challenge with the gram-negative bacteria *E. coli* (Figure 5B), gram-positive bacteria *S. aureus* (Figure 5C), and fungus *C. albicans* (Figure 5D). However, the mortality increased significantly (*p* < 0.05) in ds*AaAtg8-*treated mosquitoes after challenge with the intracellular pathogen *L. monocytogenes* compared to the ds*EGFP*-treated groups (Figure 5E). More than 80% mortality was found in the *L. monocytogenes*-challenged ds*AaAtg8*-treated adult mosquitoes at 9 days post-infection, compared with > 60% mortality in the ds*EGFP*-treated group. These findings indicates that the ds*AaAtg8*-treated adults show an increased susceptibility to infection by *L. monocytogenes*.

This experiment was repeated in adult female mosquitoes challenged with microorganisms via artificial blood feeding (Figure 6). After confirming the silencing of the *AaAtg8* transcripts in ds*AaAtg8*-treated individuals (Figure 6A), blood containing microorganisms such as *E. coli*, *S. aureus*, *L. monocytogenes*, and *C. albicans* was fed to both the ds*EGFP* (control) and ds*AaAtg8*-injected groups. The mortality of the adult mosquitoes was recorded for a duration of 9 days. Significant differences were noticed in mosquito survivability between the ds*EGFP* and ds*AaAtg8* groups after feeding with microorganisms. The silencing of *AaAtg8* resulted in a 40% reduction in the survivability of *Ae. albopictus* mosquitoes following infection by *E. coli* (Figure 6B). The survival of the ds*AaAtg8*-treated mosquitoes against the *S. aureus* challenge was 60%, which was significantly lower than that of the ds*EGFP*-treated mosquitoes (Figure 6C). The survivability against the challenge by *C. albicans* was significantly reduced (~40%) at 7 days post-challenge (Figure 6D). A decrease in survivability was also found following the challenge with *L. monocytogenes* (Figure 6E). 

These experiments reveal that *AaAtg8* is required for mosquito survival following being challenged with *E. coli*, *S. aureus, C. albicans,* and *L. monocytogenes*, which were fed to the mosquitos through blood. Hence, *AaAtg8* could play a putative role in the gut immunity of the Korean tiger mosquito *Ae. albopictus*.

## 3. Discussion

Atg8 is one of thirty-four Atgs that have been identified in yeast. Multiple orthologs of Atg8 have been identified in multicellular organisms, green plants, and some protists. In the model plant *Arabidopsis thaliana*, nine Atg8 isoforms (AtAtg8a-i) have been identified. These isoforms have been separated into two clades: one homologous to Atg8s from fungi and the other homologous to Atg8s from other animals [23]. The Atg8 proteins have been categorized into three sub-families: microtubule-associated protein 1 light chain 3 (MAP1LC3 or just LC3), γ-amino butyric acid receptor-associated protein (GABARP), and the Golgi-associated ATPase enhancer of 16 kDa (GATE-16) [24]. The members of all three sub-families have been identified in humans and other animals, including amphibians (*Xenopus tropicalis*, the African clawed frog), fish (*Danio rerio*, Zebrafish), amphioxus (*Branchiostoma floridae*), and tunicates (*Ciona savignyi*, the Pacific transparent sea squirt). Among insects, *Drosophila melanogaster* has two genes in the GABARP subfamily and *Apis mellifera* has one gene each under GABARP and LC3 sub-families [25,26]. From the transcriptome database of the coleopteran pest, *T. molitor*, two members of the Atg8 family of proteins (one belonging to the GABARP subfamily and the other to the LC3 subfamily) have been identified [5]. These findings suggest that Atg8 genes have been duplicated or lost in the process of evolution, leading to the extinction and expansion of some subfamilies and the diversification of the autophagic process) [23,24]. Atg8 has been studied extensively in the context of autophagic processes and is a commonly accepted marker of autophagy. The lipidated form of Atg8, found on the inner and outer membrane of autophagosomes, is commonly selected as a marker for autophagy induction and progression. In the autophagy mechanism, Atg8 is conjugated to the lipid PE and their association has been utilized to probe the autophagic process in eukaryotes. In the mosquito *Ae. aegypti*, Atg8 is required for autophagy in the fat body of blood-fed female mosquitoes and is involved in the prolonged production of the major yolk protein precursor vitellogenin. Therefore, it has been suggested that programmed autophagy is critical for egg maturation in mosquitoes [22]. Additionally, studies using the Atg8-PE conjugation system have confirmed that the Zika, dengue, and chikungunya viruses induce autophagy in *Ae. aegypti* Aag2 cells [27]. 

Herein, we successfully cloned the ORF region of the *AaAtg8* gene by performing a TBLASTN homology search using *TmAtg8* as a query. The conserved domain analysis predicted a ubiquitin-like fold domain in the *Aa*Atg8 protein. It is known that Atg8 homologs have a structural similarity with ubiquitin, even though there are differences in the amino acid sequences [19,28]. This domain contains important basic features for protein–protein interaction and its function is well conserved among Atg8 proteins [29]. In our study on *Aa*Atg8 proteins, we found the conserved amino acid residues responsible for interactions with Atg7 and Atg4. In fact, Atg8 interacts with several Atg proteins responsible for the regulation of autophagy, cargo recruitment, and the vacuolar fusion of autophagosomes [30,31]. The Atg8 protein of *Galleria mellonella* (*Gm*Atg8) is estimated to be 118 amino acids in length, contains a ubiquitin-like fold structure, and exposes a glycine residue at the C-terminus following cleavage by Atg4 [21]. Atg4 interaction sites were found at the carboxy-terminus of *Aa*Atg8 and are necessary for peptide cleavage and activation by the E1-like enzyme Atg7. Furthermore, Atg4 interaction sites function reversibly by de-conjugating Atg8 from the autophagic membrane [32]. The cleavage of *Aa*Atg8 by *Aa*Atg4 is expected to expose a terminal glycine (Atg8-G116), which would then be involved in the normal progression of autophagy. We consider the *Aa*Atg8 residues F_77_ and F_79_ to be critical to Atg4 recognition and cleavage. Furthermore, we consider the residue Y_49_ critical to the activation of the lipidated form of Atg8 by Atg7 and Atg3. The roles of similar residues in autophagy mechanisms have been established in yeast Atg8 [33,34]. In a previous study, *Macrobrachium nipponense* Atg4 (isoform B) silencing caused a significant downregulation of Atg8 transcripts in the brains of prawns, leading to the degeneration of brain cells [35]. Thus, it has been established that Atg4 functions as a constitutive Atg8-binding module and recruits Atg8 to the autophagosomal membranes [36,37]. Although the structural features of *A*aAtg8 have not been divulged in the present study, we predict that its ubiquitin core is stabilized by hydrogen bonds and salt bridges, as described in the *B. mori* and *G. mellonella* models [19,21].

The expression of *AaAtg8* was ubiquitous in *Ae. albopictus* tissues, including midgut, head, thorax, fat body, and Malpighian tubules, which suggests that the autophagy process is important in all tissues and critical to growth and development. In another study, *GmAtg8* expression was found to be localized in the midgut, silk gland, fat body, ovary and Malpighian tubules, and protein shifts were observed in the developmental stages [21]. Autophagy in *D. melanogaster* was also found to be active in the midgut, fat body, salivary glands, and ovary tissues [38,39]. The higher expression of *AaAtg8* during the 1 st to 4 th larval stages as compared to adult mosquitoes suggests that autophagy is a cellular organismal remodeling process required during the active phases of metamorphosis. Our results are consistent with other studies describing the role of Atg8 and the autophagy-related pathway in the developmental regulation of Lepidoptera [40,41]. Other studies have shown an increase in the levels of *GmAtg8* mRNA in the midgut during metamorphosis in the non-feeding larval stage [21]. Furthermore, the developmental expression profiles of *D. melanogaster* Atg8a and Atg8b were found to be similar to those of *GmAtg8*, yeast Atg8, and mammalian LC3 [42,43]. 

To understand the role of *AaAtg8* in host–pathogen interactions, we studied the induction of *AaAtg8* transcription in the abdominal carcass and midgut of the adult *Ae. albopictus* after being challenged by various microorganisms via injection or artificial blood feeding. Our studies revealed the ability of the intracellular pathogen, *L. monocytogenes,* to induce the transcription of *AaAtg8* in sugar- and blood-fed mosquitoes. In fact, *L. monocytogenes* is known to orchestrate autophagy machinery in a way that, in turn, protects the host from the invading pathogen. It is also known that *L. monocytogenes* can circumvent the autophagy machinery and promote phagocytosis [44]. Other bacteria (including *Streptococcus pyogenes, Mycobacterium tuberculosis*, *Shigella flexneri, Salmonella typhimurium,* and *Legionella pneumophila*) elicit a known reaction in the autophagic machinery, and some are capable of evading cellular responses for increased survival within the host [45]. It is suggested that *L. monocytogenes* can escape the phagosome by secreting both broad-range phosphatidylcholine, which helps to form pores in the autophagosomal membrane, and listeriolysin O (LLO) [46,47]. *TmAtg8*-silenced larvae were shown to be susceptible to a *Listeria* infection in hemocytes, suggesting that Atg8 plays a role in clearing *Listeria* from *T. molitor* [5]. Our investigations also revealed that *E. coli* was able to strongly induce *AaAtg8* transcription in the midgut of mosquitoes that were fed microorganisms in blood. Studies in the *Drosophila* model have indicated that *E. coli* infections can encourage the autophagic machinery of the host to clear the pathogen [48]. Other pathogens used in the present study, such as the gram-positive *S. aureus* and the fungus *C. albicans,* failed to significantly induce *AaAtg8* transcription. 

To functionally characterize *AaAtg8* and understand its utility in clearing microorganisms from the host, we performed microbial challenge studies in *AaAtg8* gene knockdown mosquito models. In two separate sets of experiments, we conducted a survival assay involving adult *Ae. albopictus* mosquitoes that had been challenged with microorganisms (*E. coli*, *S. aureus*, *C. albicans*, and *L. monocytogenes*) via systemic injection or through blood feeding. In models with knockdown efficiency of more than 95%, we compared the survivability of ds*AaAtg8*-treated mosquitoes to ds*EGFP*-treated (control) mosquitoes. Following systemic injection of *L. monocytogenes*, survivability of the ds*AaAtg8*-treated mosquitoes was significantly reduced compared to the control group. This response was unique to challenge with *L. monocytogenes*, silencing of *AaAtg8* had no significant effect on the survivability of the adult mosquitoes in response to challenge with *E. coli*, *S. aureus*, and *C. albicans*. Infection of *T. molitor* hemocytes by *L. monocytogenes* has been reported at 6 h post-infection using the autophagosome-specific probe Cyto-ID green dye and the endo-lysosomal probe LysoTracker Red DND-99. In addition, *TmAtg8* was shown to increase clearance of the intracellular pathogen *L. monocytogenes* from hemocytes via programmed autophagic mechanisms [5]. In *Drosophila*, *L. monocytogenes* invades the cytosol of larval hemocytes, is recognized by host PGRP-LE, and is eventually degraded by the autophagic machinery of the host [49]. Knockdown of *T. molitor* PGRP-LE has also been shown to reduce survivability of the host larvae when faced with infection by *L. monocytogenes*, suggesting the necessity of PGRP-LE for induction of autophagy-mediated control of the pathogenic infection [50]. Another study has confirmed the activation of the phenoloxidase system preceding *L. monocytogenes* pathogenesis in the larvae of *G. mellonella* [51]. In the present study, we observed a significant decrease in the survivability of *Ae. albopictus* adult mosquitoes after challenge by microorganisms such as *E. coli*, *S. aureus*, *C. albicans*, and *L. monocytogenes* by blood feeding. This suggests that *AaAtg8* is required for pathogenic surveillance in the host and improves gut-specific immunity. Generally, gut tissue is a candidate for studies on bacterial infections that lead to the activation of both humoral and cell-mediated immunity [52]. More specifically, the activation of the dual oxidase (DUOX) pathway in *Drosophila* following infection with the enteric pathogen, *Erwinia carotovora* subspecies *carotovora*, has been shown previously [53]. In the case of mosquitoes, gut immunity has been explored to a reasonable level in regard to viral infection [54,55]. This study advances this understanding by relating gut immunity to the autophagy marker protein Atg8 and describing one method by which mosquitoes respond to bacterial and fungal infections.

## 4. Materials and Methods 

### 4.1. Mosquito Rearing 

Experiments were conducted with the Korean strain of *Ae. albopictus*. The mosquito larvae were obtained from the Korea Centers for Disease Control and Prevention (KCDC) and reared at 27 ± 1 °C and 60% ± 5% relative humidity under a 12:12 h light: dark cycle. While larval mosquitoes were grown in tap water and fed with the tropical fish feed TetraMin powder, adult mosquitoes were reared on a 10% sugar solution and fed on defibrinated sheep blood (Biozoa Biological Supply Co., Seoul, Korea) under the same environmental conditions.

### 4.2. Microbial Cultures and Infection of Mosquitoes 

The gram-negative bacterium *E. coli* (strain K12); gram-positive bacteria *S. aureus* (strain RN4220); *L. monocytogenes* (strain ATCC7644); and the fungus *C. albicans* were used for the immune challenge experiments. The *E. coli* and *S. aureus* were cultivated in a Luria–Bertani (LB) broth. The *L. monocytogenes* and *C. albicans* were cultured in a brain–heart infusion (BHI) broth and Sabouraud dextrose broth, respectively. All the microbiological culture media were procured from MB Cell, Korea. The microbial cells were cultured overnight at 37 °C and harvested, washed, and suspended in a 1× phosphate-buffered saline solution (PBS; pH 7.0). The washed cells were centrifuged at 1146× *g* for 10 min. The concentration of the washed microorganisms was measured at 600 nm (OD_600_) using a spectrophotometer (Eppendorf, Germany). Finally, the cells were diluted in PBS to reach concentrations of 1 × 10^5^ cells/μL for *E. coli* and *S. aureus*, 2.5 × 10^4^ cells/μL for *C. albicans*, and 5 × 10^5^ cells/μL for *L. monocytogenes* to be used in the immune challenge studies.

### 4.3. Identification and in Silico Analysis of AaAtg8

The *AaAtg8* gene sequence was retrieved by performing a local TBLASTN analysis using the *Tenebrio molitor* Atg8 amino acid sequence (AJE26300.1) as the query and an unpublished *Ae. albopictus* nucleotide database, generated from an RNA sequencing analysis, as the subject. To confirm the full-length open reading frame (ORF) sequence of *AaAtg8*, gene specific primers (as provided in Table 1) were designed using the program primer3 (http://bioinfo.ut.ee/primer3-0.4.0/). The target gene was amplified by the AccuPower^®^ Pfu PCR PreMix (Bioneer, Daejeon, South Korea) on a MyGenie 96 thermal block (Bioneer, Daejeon, South Korea). The PCR product was cloned into the T-blunt vector using the T-blunt^TM^ PCR Cloning Kit (Solgent Company, South Korea), transformed into competent *E. coli* (DH5α) cells, and sequenced. For domain analysis and residue functions, the expected amino acid sequence of *AaAtg8* was subjected to InterProScan (https://www.ebi.ac.uk/interpro/search/sequence-search) and BLASTP analyses. In addition, we performed a pairwise alignment of the *AaAtg8* gene of the Korean and Chinese strains (XP_001652571.1) of *Ae. albopictus* at the nucleotide and amino acid sequence levels using the Clustal X2.1 program [56]. 

### 4.4. Expression and Induction Patterns of AaAtg8

The *AaAtg8* mRNA expression was determined in the developmental stages of *Ae. albopictus* such as egg (EG), 1 st–4 th instar larvae (Lv1–Lv4), prepupae (PP), pupae (P), adult male mosquitos (AM), adult sugar-fed female (ASF), and adult blood-fed female mosquitoes (ABF). The *AaAtg8* mRNA expression was also measured in different tissues, including the head (HD), thorax (TX), midgut (MG), fat body (FB), and Malpighian tubules (MT) of sugar- and blood-fed mosquitoes, as well as the ovary (OV) tissue of blood-fed mosquitoes. 

To investigate the induction patterns of *AaAtg8* mRNA expression in response to microbial infection, *E. coli* (2 × 10^4^ cells/mosquito), *S. aureus* (2 × 10^4^ cells/mosquito), *C. albicans* (5 × 10^3^ cells/mosquito), and *L. monocytogenes* (1 × 10^5^ cells/mosquito) were injected into sugar-fed (3-day-old adult female mosquitoes) and blood-fed mosquitoes (5-day-old adult female mosquitoes). In addition, microorganisms in equivalent concentrations were used to infect 5-day-old adult female mosquitoes by artificial feeding with sheep blood. Abdominal carcass and midgut samples were collected at 3, 6, 9, and 12 h post-infection.

The total RNA was isolated from the samples by the Clear-S^®^ total RNA extraction kit (Invirustech Co., Gwangju, Korea) according to the manufacturer’s instructions. Immediately, 2 μg of Total RNAs were used as the template to synthesize cDNA using an Oligo-(dT)_12–18_ primer and AccuPower^®^ RT PreMix (Bioneer, Daejeon, Korea) according to manufacturer’s instructions. 

The relative expression level of *AaAtg8* mRNA was investigated by performing quantitative real-time PCR (qRT-PCR) using the AccuPower^®^ 2 × GreenStar™ qPCR Master Mix (Bioneer, Daejeon, Korea), with synthesized cDNAs and *AaAtg8* gene specific primers designed using the Primer3Plus program (http://primer3plus.com/cgi-bin/dev/primer3plus.cgi) (Table 1). The PCR cycling conditions included an initial denaturation at 95 °C for 5 min, followed by 40 cycles of denaturation at 95 °C for 15 s, and annealing and extension at 60 °C for 30 s. The qRT-PCR assays were performed on an AriaMx Real-Time PCR System (Agilent Technologies, Santa Clara, CA, USA), and the results were analyzed using the AriaMx Real-Time PCR software. The 2^−*∆∆C*t^ method [57] was employed to analyze *AaAtg8* mRNA expression levels. The mRNA expression levels were normalized to those of *Ae. albopictus ribosomal protein S*6 (*AaRps6*), which acted as an internal control. The results represent the mean ± Standard Error of three biological replicates.

### 4.5. dsRNA Synthesis and AaAtg8 Silencing in Female Mosquitoes

To synthesize the dsRNA fragment of the *AaAtg8* gene for RNAi-based silencing experiments, a dsDNA fragment of *AaAtg8* was amplified by PCR using gene-specific primers conjugated with T7 promoter sequences (Table 1). The primers were designed using SnapDragon (https://www.flyrnai.org/cgi-bin/RNAi_find_primers.pl) to prevent any cross-silencing effects. The PCR was carried out using the AccuPower^®^ Pfu PCR PreMix (Bioneer, Daejeon, Korea) with a cDNA template and specific primers for the *AaAtg8* gene (Table 1). The PCR cycling conditions were identical to those described in Section 4.4. The PCR products were purified with the AccuPrep^®^ PCR Purification Kit (Bioneer, Daejeon, Korea) and used to synthesize the dsRNA using the EZ™ T7 High Yield in Vitro transcription Kit (Enzynomics, Daejeon, Korea). This process was also used to synthesize the dsRNA for the Enhanced Green Fluorescent protein (ds*EGFP*) for use as a negative control. The dsRNA product was purified by the PCI (phenol:chloroform:isopropyl alcohol mixture) method and precipitated with 5M ammonium acetate and ethanol. The ds*EGFP* and ds*AaAtg8* were stored at −20°C until further use. *AaAtg8* knockdown models were generated by injecting 200 ng of ds*EGFP* or ds*AaAtg8* into *Ae. albopictus* (3-day-old sugar-fed) female mosquitoes. 

### 4.6. Effect of AaAtg8 Silencing on Survival of Mosquitoes after Inoculation with Microorganisms

The significance of *AaAtg8* in the survival of *Ae. albopictus* mosquitoes were assessed in dsRNA-injected groups following both microbial injection and ingestion (*E. coli*, *S. aureus*, *C. albicans* and *L. monocytogenes*). The dead mosquitoes were counted daily for up to 9 days post-infection. The experiments were conducted in three biological replicates (*n* > 8). The results for the mortality assay were presented as per the Kaplan–Meier plot method. 

### 4.7. Statistical Analysis

All experiments were carried out in triplicate and the data was subjected to a one-way analysis of variance (ANOVA) and Tukey’s multiple range tests (*p* < 0.05).

## 5. Conclusions

This study reports for the first time the cloning, expression analysis, and functional characterization of an Atg8 homolog in the Korean strain of the Asian tiger mosquito, *Ae. albopictus*. While the intracellular gram-positive bacterium *L. monocytogenes* induced *AaAtg8* translation in systemically infected blood-fed mosquitoes, the gram-negative bacterium *E. coli* significantly induced transcription in mosquitoes that were challenged by being fed microorganisms in blood. Interestingly, the knockdown of *AaAtg8* led to an increased susceptibility of adult mosquitoes to a *L. monocytogenes* infection after systemic injection. In contrast, microbe feeding made the adult mosquitoes susceptible to bacteria such as *E. coli*, *S. aureus*, *L. monocytogenes* and the fungus *C. albicans*. This therefore suggests that *AaAtg8* is required to confer antimicrobial immunity in the gut of the Asian tiger mosquito *Ae. albopictus* (Figure 7).

## Figures and Tables

**Figure 1 ijms-21-02944-f001:**
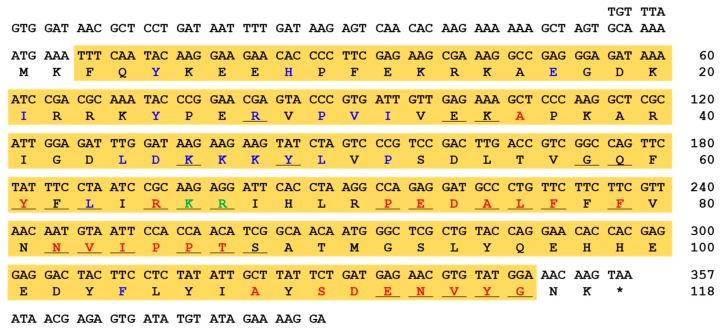
Nucleotide and encoded polypeptide sequence of *AaAtg8*. The asterisk denotes stop codon. The autophagy protein Atg8 ubiquitin-like domain is shaded in yellow. The Atg7 interaction sites (thirty amino acid residues) are underlined. Further, the Atg4 interaction sites (twenty four amino acid residues) are shown in red text. The peptide binding site (nineteen amino acid residues) and key conserved lysine residues (K66 and R67) are highlighted in blue and green text, respectively.

**Figure 2 ijms-21-02944-f002:**
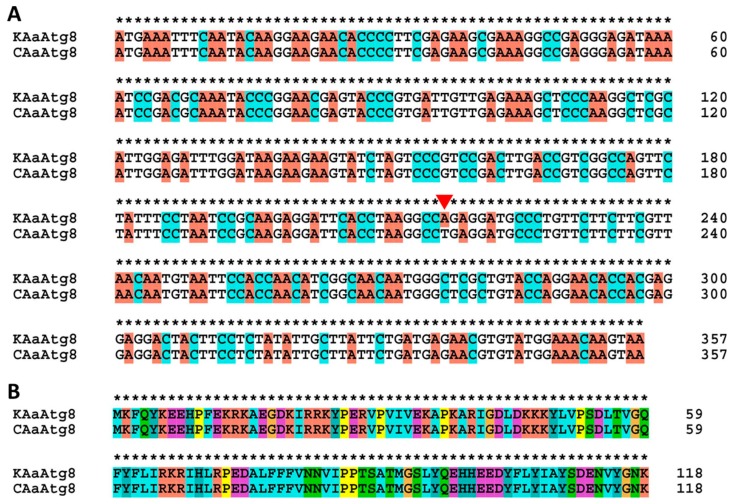
Comparative analysis of the *AaAtg8* gene identified from the *Ae. albopictus* Korean (*KAaAtg8*) and Chinese strains (*CAaAtg8*) at the nucleotide (**A**) and amino acid sequence (**B**) level. The pairwise alignment results show a single nucleotide mismatch (marked by a red triangle). However, the mismatch of A_216_ in *KAaAtg8* to T_216_
*CAaAtg8* does not alter the expected protein sequence.

**Figure 3 ijms-21-02944-f003:**
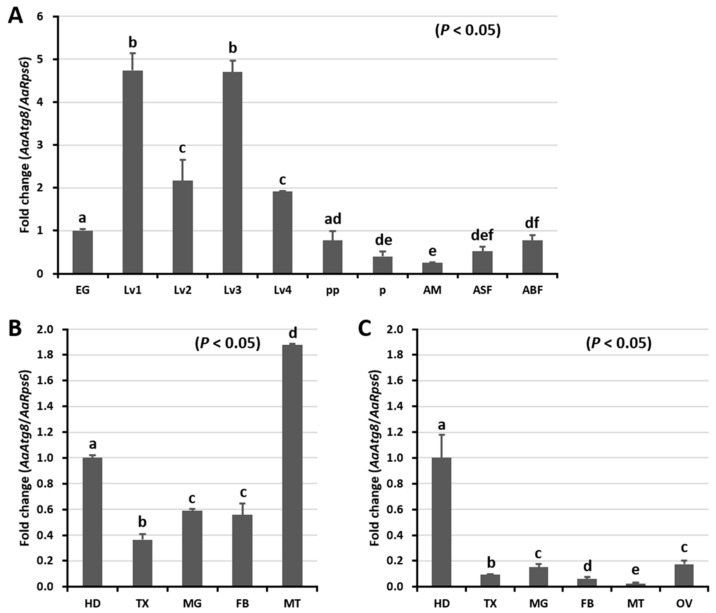
Developmental and tissue-specific expression patterns of *AaAtg8* mRNA. (**A**) Developmental expression patterns of *AaAtg8* mRNA. EG, eggs; Lv1, 1st instar larvae; Lv2, 2nd instar larvae; Lv3, 3rd instar larvae; Lv4, 4th instar larvae; PP, prepupae; P, pupae; AM, adult males; ASF, adult sugar-fed females; ABF, adult blood-fed females. Tissue-specific expression profiles of *AaAtg8* in adult sugar-fed (**B**) and blood-fed (**C**) female mosquitoes. HD, head; TX, thorax; MG, midgut; FB, fat body; MT, Malpighian tubules; OV, ovary. The *Ae. albopictus ribosomal protein S6* (*AaRps6*) mRNA expression was used as an endogenous control. Data presented show the mean ± standard error. Different letters above the bars show significant differences (*p* < 0.05) between groups (*n* = 3).

**Figure 4 ijms-21-02944-f004:**
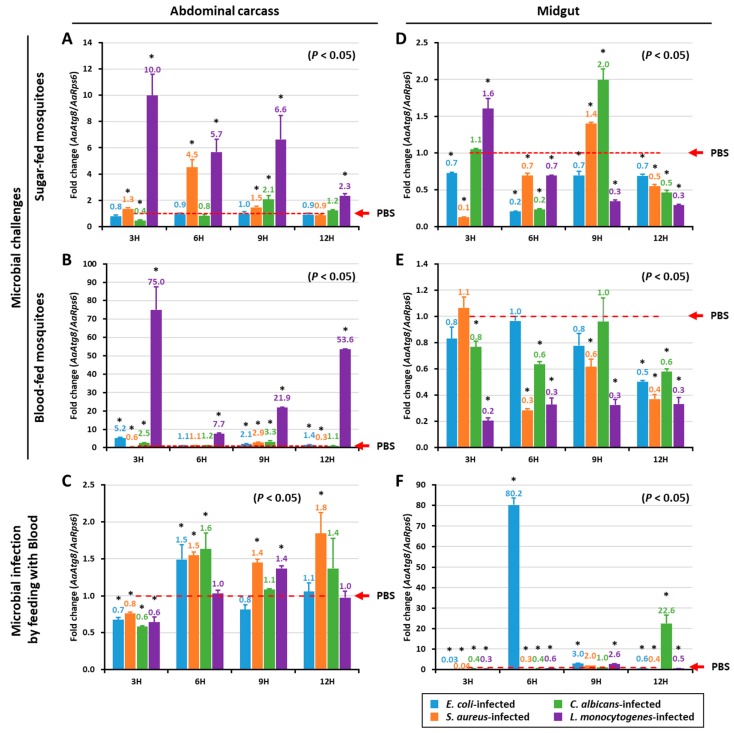
Temporal expression of *AaAtg8* mRNA in response to microorganism challenge by injection and feeding. *AaAtg8* mRNA levels are expressed as fold changes and were measured in the abdominal carcass and midgut tissue. While the expression profiles of *AaAtg8* mRNA in the abdominal carcass of sugar-fed and blood-fed mosquitoes after the microorganisms challenge are shown in (**A**,**B**), the expression profiles of *AaAtg8* mRNA in the midgut tissue of sugar-fed and blood-fed mosquitoes after microorganisms challenge are shown in (**D**,**E**). The expression profiles of *AaAtg8* mRNA after microbial infection by feeding with blood in the abdominal carcass and midgut are shown in (**C**,**F**), respectively. The fold change in expression levels in groups infected with microorganisms was compared with the phosphate-buffered saline solution (PBS)-injected (control) group. The *Ae. albopictus ribosomal protein S6* (*AaRps6*) mRNA expression was used as an endogenous control. Results of the triplicate experiments are shown with standard errors. An asterisk indicates significance at a 95% confidence level (Statistical Analysis Software, ANOVA).

**Figure 5 ijms-21-02944-f005:**
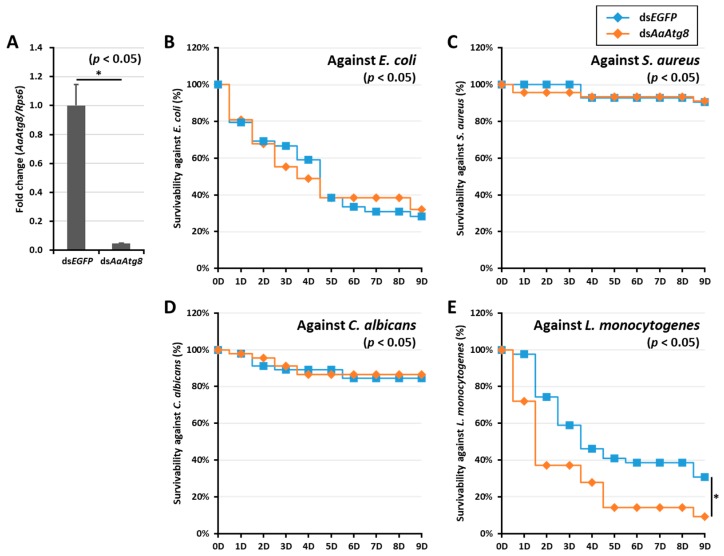
Effects of *AaAtg8* mRNA silencing on microorganism infection in sugar-fed *Ae. albopictus* mosquitoes. (**A**) RNAi efficiency of *AaAtg8* mRNA in ds*AaAtg8*-injected adult mosquito group (*n* > 24) compared with ds*EGFP*-injected adult mosquito group (*n* > 24). The ds*AaAtg8* and ds*EGFP* were injected at a concentration of 200 ng/mosquito. The survival rate of ds*AaAtg8*-treated adult mosquitoes is shown against that of those treated with *E. coli* (**B**), *S. aureus* (**C**), *C. albicans* (**D**) and *L. monocytogenes* (**E**). The mosquito survivability was investigated for 9 days. The survivability curve is represented by a Kaplan–Meier plot and asterisks (*) indicate a significant difference at 95% confidence levels.

**Figure 6 ijms-21-02944-f006:**
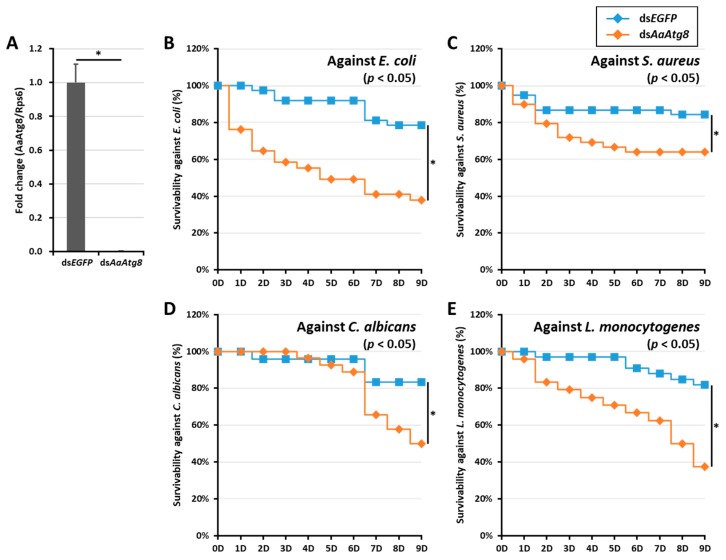
Effects of *AaAtg8* mRNA silencing on mosquito survivability against microbial feeding. (**A**) RNAi efficiency of *AaAtg8* mRNA in the ds*AaAtg8*-injected adult mosquito group (n > 24) compared with the ds*EGFP*-treated group. Four different pathogens, *E. coli* (**B**), *S. aureus* (**C**), *C. albicans* (**D**), and *L. monocytogenes* (**E**), were fed in blood to the ds*AaAtg8*-treated adult mosquito group and the survivability was investigated for 9 days. Data represent the average of three independent biological replications. The survivability curve is represented by a Kaplan–Meier plot and an asterisk (*) indicates a significant difference at 95% confidence levels.

**Figure 7 ijms-21-02944-f007:**
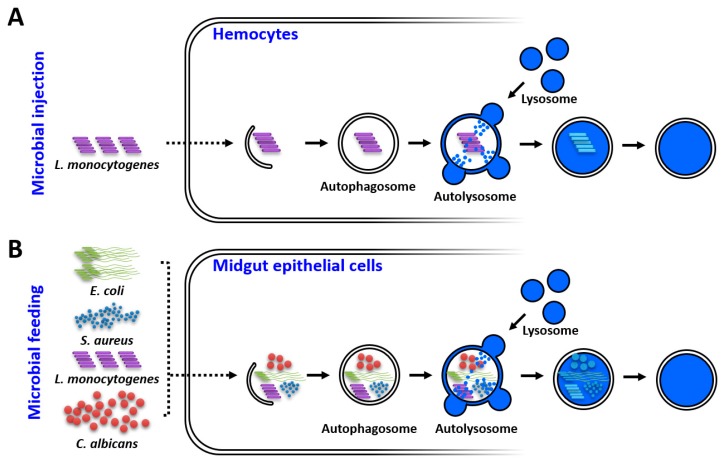
Graphical illustration of *AaAtg8* functions in response to microbial infections by injection (**A**) and ingestion (**B**) in *Ae. albopictus*.

**Table 1 ijms-21-02944-t001:** Primer sequences used in the study.

Name	Primer Sequences
AaAtg8-cloning-452bp-Fw AaAtg8-cloning-452bp-Rv	5′-TGTTTAGTGGATAACGCTCCTG-3′ 5′-TCCTTTTCTATACATATCACTCTCGTT-3′
AaAtg8-qPCR-Fw AaAtg8-qPCR-Rv	5′-TACAAGGAAGAACACCCCTTCG-3′ 5′-AATCTCCAATGCGAGCCTTG-3′
AaAtg8-dsRNA-296bp-Fw AaAtg8-dsRNA-296bp-Rv	5′-TAATACGACTCACTATAGGGT AAAGGCCGAGGGAGATAAAA-3′ 5′-TAATACGACTCACTATAGGGT GGTGTTCCTGGTACAGCGAG-3′
EGFP-dsRNA-Fw EGFP-dsRNA-Rv	5′-TAATACGACTCACTATAGGGT ACGTAAACGGCCACAAGTTC-3′ 5′-TAATACGACTCACTATAGGGT TGCTCAGGTAGTGTTGTCG-3′

Underline: T7 promoter sequences.

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
