# Peer review of "Aedes albopictus Autophagy-Related Gene 8 (AaAtg8) Is Required to Confer Anti-Bacterial Gut Immunity"

_ijms, 2020, doi:10.3390/ijms21082944_

Round 1

Reviewer 1 Report

The manuscript „Aedes albopictus autophagy-related gene 8 (AaAtg8) is required to confer anti-bacerial gut immunity“ by the authors Kim et al. provides very interesting insights into the role of the autophagy related protein 8 (Atg8) of the Asian tiger mosquito Aedes albopictus which is involved in the transmission of several important vector-borne human virus diseases. The manuscript is well written and experiments and conclusions are coherent.

It remains however unclear to me why Listeria monocytogenes and the other bacterial/fungal microorganisms were chosen as a model pathogens for this study. Especially since this insect vector is rather relevant for transmitting viral diseases one would assume that the role of Atg8 during virus defence within the vector is even more interesting. Do infections with the microorganism from your study naturally occur in A. albopictus?  Please mention the rationale for the choice of these microorganisms in the introduction.

The authors silenced Atg8 by mRNA silencing and could clearly show that Atg8 plays a role during pathogen infection. Is there any hypothesis about the the role of Atg8 in the immune system on the molecular level? How about the bacterial/fungal load in the insects during the course of infection is there a difference in pathogen eradication between the AaAtg8 silenced mosquities and the control group? Re-isolation and/or quantification of the microorgansims initially used for the challenge/infection should be feasible from the insects and would provide interesting insights about the (potential) role of AaAtg8 in pathogen clearance. This could be done by plating insect lysates from different time points on pathogen-selective media or by pathogen specific qPCR normalized to host DNA.

General comments

In the text it is sometimes difficult to understand the methodological differences between infection, challenge and injection of microorganisms. That could be written in a more comprehensible manner.

line

33          better write „“…challenged by injection or blood-feeding of…“

34          I assume the effect was seen in mosquitos in which Aatg8 was silenced? Please rewrite the sentence

58          delete „known“ in this line

101-5    better avoid the long list of amino acids in the text. Refer to Fig. 1.

Why is in Fig. 3C the bar for ovaries missing? Please use the same y-axis scale in Fig. 3B and C to allow better visual comparison.

153-158               The part beginning with „In the midgut tissue…“ is not clear to me. What ist he difference between sugar- and blood fed mosquitoes challenged with microbes and mosquitoes challenged with microbes by artificial blood feeding“. In other words what is the experimental difference between Fig.4 B/E and C/F? In the legend of Fig. 4 it is a bit better explained but still unclear with regards to the terms infection, injection, challenge and feeding. Please rewrite the text to make it easier understandable.

Author Response

The manuscript „Aedes albopictus autophagy-related gene 8 (AaAtg8) is required to confer anti-bacterial gut immunity“ by the authors Kim et al. provides very interesting insights into the role of the autophagy related protein 8 (Atg8) of the Asian tiger mosquito Aedes albopictus which is involved in the transmission of several important vector-borne human virus diseases. The manuscript is well written and experiments and conclusions are coherent.

Author’s response: Indeed a pleasure to hear positive comments from the reviewer regarding the manuscript.

It remains however unclear to me why Listeria monocytogenes and the other bacterial/fungal microorganisms were chosen as a model pathogens for this study. Especially since this insect vector is rather relevant for transmitting viral diseases one would assume that the role of Atg8 during virus defense within the vector is even more interesting. Do infections with the microorganism from your study naturally occur in A. albopictus?  Please mention the rationale for the choice of these microorganisms in the introduction.

Author’s response: The authors of the manuscript are over-whelmed with the suggestive guidance. Initially when we designed the plan of this study, we hypothesized that bacteria could be used as a mosquitocide, and the means to improve the efficiency of this containment. To study this hypothesis, we were interested to note the immune pathways related to microbial infection. Hence, we have selected microbes (bacteria and fungus) as model pathogens, and not viruses. In addition, it was important to understand whether autophagy which has an important role in anti-viral immune responses (below references), could be involved in anti-bacterial immunity.

Echavarria-Consuegra L, Smit JM, Reggiori F. Role of autophagy during the replication and pathogenesis of common mosquito-borne flavi- and alphaviruses. Open Biol. 2019 Mar 29;9(3):190009. doi: 10.1098/rsob.190009. PMID: 30862253; PMCID: PMC6451359.

Doug E. Brackney, Maria A. Correa. A limited role for autophagy during arbovirus infection of mosquito cells. BioRxiv preprint doi: https://doi.org/10.1101/760728

Further, mosquitoes such as the Aedes species are naturally infected by several microbes in various environmental conditions (see below reference). Dada N, Jumas-Bilak E, Manguin S, Seidu R, Stenström TA, Overgaard HJ. Comparative assessment of the bacterial communities associated with Aedes aegypti larvae and water from domestic water storage containers. Parasit Vectors. 2014 Aug 24;7: 391. doi: 10.1186/1756-3305-7-391. PMID: 25151134; PMCID: PMC4156648.

Bennett KL, Gomez-Martinez C, Chin Y, Saltonstall K, McMillan WO, Rovira JR, Loaiza JR. Dynamic and diversity of bacteria associated with the disease vectors Aedes aegypti and Aedes albopictus. Scientific Reports 9, 12160 (2019).

The authors silenced Atg8 by mRNA silencing and could clearly show that Atg8 plays a role during pathogen infection. Is there any hypothesis about the role of Atg8 in the immune system on the molecular level? How about the bacterial/fungal load in the insects during the course of infection is there a difference in pathogen eradication between the AaAtg8 silenced mosquitoes and the control group? Re-isolation and/or quantification of the microorganisms initially used for the challenge/infection should be feasible from the insects and would provide interesting insights about the (potential) role of AaAtg8 in pathogen clearance. This could be done by plating insect lysates from different time points on pathogen-selective media or by pathogen specific qPCR normalized to host DNA.

Author’s Response:

As per recent publications, autophagy has various functions. Especially, it plays critical role in clearance of non-used or broken components in the cells as well as non-self components such as viral, bacterial or fungal pathogens (see below references).

Subauste CS. Autophagy as an antimicrobial strategy. Expert Rev Anti Infect Ther. 2009 Aug;7(6):743-52. doi: 10.1586/eri.09.41. PMID: 19681702; PMCID: PMC3280690.

Ligia C.Gomes, Ivan Dikic. Autophagy in Antimicrobial Immunity. Molecular Cell. Volume 54, Issue 2, 24 April 2014, Pages 224-233 https://doi.org/10.1016/j.molcel.2014.03.009

Boyle K.B., Randow F. (2019) Measuring Antibacterial Autophagy. In: Ktistakis N., Florey O. (eds) Autophagy. Methods in Molecular Biology, vol. 1880. Humana Press, New York, NY

Additionally, in the Tenebrio molitor model, we identified and characterized the immune function of Atg8 in response to bacterial infection (see below reference). Atg8 plays dual role in autophagosome formation process by coupling between selective incorporation of autophagy-cargo and promoting autophagosome membrane expansion and closure.

Tindwa H, Jo YH, Patnaik BB, Lee YS, Kang SS, Han YS (2015) Molecular cloning and characterization of autophagy-related gene TmATG8 in Listeria-invaded hemocytes of Tenebrio molitor. Dev Comp Immunol 51: 88-98

We apologize that the experiments involving plating insect lysates from different time-points cannot be done this time around, as sample preparations including mosquito rearing and conducting the experiments takes a minimum of one month. We would definitely incorporate the suggestions in future experiments.

General comments

In the text it is sometimes difficult to understand the methodological differences between infection, challenge and injection of microorganisms. That could be written in a more comprehensible manner.

line

33          better write „“…challenged by injection or blood-feeding of…“

34          I assume the effect was seen in mosquitos in which Aatg8 was silenced? Please rewrite the sentence

58          delete „known “in this line

101-5    better avoid the long list of amino acids in the text. Refer to Fig. 1.

Author’s response: All the above comments have been taken care in the revised manuscript.

Why is in Fig. 3C the bar for ovaries missing? Please use the same y-axis scale in Fig. 3B and C to allow better visual comparison.

Author’s response: The revision has been made as shown below-               

153-158               The part beginning with „In the midgut tissue…“ is not clear to me. What is he difference between sugar- and blood fed mosquitoes challenged with microbes and mosquitoes challenged with microbes by artificial blood feeding“. In other words what is the experimental difference between Fig.4 B/E and C/F? In the legend of Fig. 4 it is a bit better explained but still unclear with regards to the terms infection, injection, challenge and feeding. Please rewrite the text to make it easier understandable.

Author’s response: Necessary revisions have been made in the manuscript

Reviewer 2 Report

General assessment and comments:

In the submitted manuscript, Kim et al addressed the antimicrobial function of autophagy-related protein 8 (AaAtg8) in mosquito Aedes albopictus. The authors first clone the gene from Aedes albopictus and performed sequence analysis. In addition, the authors checked the tissue expression pattern of AaAtg8 mRNA in Aedes albopictus. Using RNAi, the authors investigated the role of AaAtg8 in antimicrobial responses in Aedes albopictus.

This study addresses the antimicrobial function of autophagy-related protein 8 (AaAtg8) in mosquito, which will provide helpful information to the readers in the field who are interested in this topic. Overall, the manuscript is well written. Here are some suggestions for authors to consider. 

1. Line 141, change “after immune stimulation”  to “after microbial challenge”.

2. Line 175-176, better clarify the challenge route.

Author Response

In the submitted manuscript, Kim et al addressed the antimicrobial function of autophagy-related protein 8 (AaAtg8) in mosquito Aedes albopictus. The authors first clone the gene from Aedes albopictus and performed sequence analysis. In addition, the authors checked the tissue expression pattern of AaAtg8 mRNA in Aedes albopictus. Using RNAi, the authors investigated the role of AaAtg8 in antimicrobial responses in Aedes albopictus.

This study addresses the antimicrobial function of autophagy-related protein 8 (AaAtg8) in mosquito, which will provide helpful information to the readers in the field who are interested in this topic. Overall, the manuscript is well written. Here are some suggestions for authors to consider. 

Author’s response: We thank the reviewer for positive comments.

  1. Line 141, change “after immune stimulation”to “after microbial challenge”.
  2. Line 175-176, better clarify the challenge route.

Author’s response: The necessary corrections are complete from the author’s side.

Round 2

Reviewer 1 Report

I agree, it is not easily feasible to repeat all experiments to provide the pathogen quantification as requested. A recommendation from my side for the next study is to keep some insects of each time point, freeze them at -80°C (just plain, without ethanol or anything else is the best) and do the DNA purification and qPCR with pathogen specific primers after the other analyses are completed. All other comments where adressed to my satisfaction.